# Plant Coping with Cold Stress: Molecular and Physiological Adaptive Mechanisms with Future Perspectives

**DOI:** 10.3390/cells14020110

**Published:** 2025-01-13

**Authors:** Yan Feng, Zengqiang Li, Xiangjun Kong, Aziz Khan, Najeeb Ullah, Xin Zhang

**Affiliations:** 1Henan Collaborative Innovation Centre of Modern Biological Breeding, Henan Institute of Science and Technology, Xinxiang 453003, China; fengyan02020@163.com (Y.F.); lizengqiang2020@163.com (Z.L.); kongxiangjun201010@163.com (X.K.); 2State Key Laboratory of Herbage Improvement and Grassland Agroecosystems, College of Ecology, Lanzhou University, Lanzhou 730000, China; aziz.hzau@gmail.com; 3Department of Agronomy, College of Agriculture, Shandong Agriculture University, Tai’an 271018, China; 4Agricultural Research Station, Office of VP for Research & Graduate Studies, Qatar University, Doha 2713, Qatar; nullah@qu.edu.qa

**Keywords:** chilling, low temperature, physiological biochemistry, phytohormone, signaling pathway, transcription factors

## Abstract

Cold stress strongly hinders plant growth and development. However, the molecular and physiological adaptive mechanisms of cold stress tolerance in plants are not well understood. Plants adopt several morpho-physiological changes to withstand cold stress. Plants have evolved various strategies to cope with cold stress. These strategies included changes in cellular membranes and chloroplast structure, regulating cold signals related to phytohormones and plant growth regulators (ABA, JA, GA, IAA, SA, BR, ET, CTK, and MET), reactive oxygen species (ROS), protein kinases, and inorganic ions. This review summarizes the mechanisms of how plants respond to cold stress, covering four main signal transduction pathways, including the abscisic acid (ABA) signal transduction pathway, Ca^2+^ signal transduction pathway, ROS signal transduction pathway, and mitogen-activated protein kinase (MAPK/MPK) cascade pathway. Some transcription factors, such as AP2/ERF, MYB, WRKY, NAC, and bZIP, not only act as calmodulin-binding proteins during cold perception but can also play important roles in the downstream chilling-signaling pathway. This review also highlights the analysis of those transcription factors such as bHLH, especially bHLH-type transcription factors ICE, and discusses their functions as phytohormone-responsive elements binding proteins in the promoter region under cold stress. In addition, a theoretical framework outlining plant responses to cold stress tolerance has been proposed. This theory aims to guide future research directions and inform agricultural production practices, ultimately enhancing crop resilience to cold stress.

## 1. Introduction

Crop yields are negatively affected by cold stress, including chilling (0~12 °C) and freezing stress (≤0 °C) resulting in constrained sowing time, damage to plant tissues, and stunted plant growth [1]. Tolerance to cold stress in plants can be divided into chill-susceptible (<12 °C plants will be damaged), chill-tolerant but freezing susceptible (0~12 °C plants can adapt, but once the tissues are frozen, the cells will be damaged), and freeze-tolerant (survival < 0 °C) [2]. Cold stress seriously threatens global crop productivity and yield reduction in temperate climates [3]. The influence of cold stress depends on the status of plant cellular morphology (e.g., membranes and organelles) [4], biochemistry (e.g., enzyme activity) [5], physiological functioning (e.g., gas exchange and water management) [6], and phenology (e.g., developmental stages) [7]. The physio-biochemical adaptive mechanisms of plants have been widely discussed, but less attention has been paid to molecular adaptive mechanisms. Thus, plant responses to cold stress are complex and multidimensional, involving signal perception and transduction.

Plants respond to cold stress through a series of physiological adaptations, including regulation of phytohormones (e.g., abscisic acid, brassinosteroids, and jasmonic acid) [8], osmotic substances (e.g., soluble sugar and betaine) [9], and inorganic ions (Ca^2+^) [10]. Plants may acquire cold resistance through cold acclimation [11], in which exposure of plants to low temperatures (0~5 °C lower than threshold temperature and above 0 °C) for a certain period (usually several days or weeks) [12] allows them to adapt to cold stress through phytohormone, antioxidants, osmotic regulation substances, and inorganic ions. They act as intercellular signals linking environmental stimuli to plant responses, and these signal transduction pathways are often accomplished through extensive transcriptional regulation. Understanding the changes in morphological, physiological, and molecular metabolism of plants in response to cold stress is critical for cold-tolerant breeding.

## 2. Morphological Changes in Response to Cold Stress in Plants

Cold tolerance is an environmental adaptation characteristic resulting from various changes in morphological structure, gene expression, and protein expression caused by the interaction between genes and the environment. In this section, we first consider cold stress in the context of the whole plant’s developmental stages, and then highlight the need to identify and evaluate the varying cold tolerance of different plants.

All stages of plant development, from seed germination, flowering, and fruiting to dormancy, are impacted by cold stress. Various levels of cold stress can influence the same plant tissue at distinct growth stages. There are many plants for which cold damage classification is based on leaf damage [13]. Cold stress leads to insufficient leaf [14] and root [15] development (Table 1). Through principal component analysis, root characteristics may aid in identifying corn (*Zea mays* L.) hybrids that exhibit cold tolerance [16]. In one study, 15 hybrids (45%) had cumulative root length greater than the average total root length under extreme cold treatment [16]. Kinematic analysis of diurnal growth rates in control and cold-treated corn leaves from germination until the completion of leaf 4 expansion showed that cold nights had an impact on both cell cycle time (+65%) and cell yield (−22%); meanwhile, the size of mature epidermal cells was unaffected [17]. This finding contrasts with the common belief that the reduction in growth caused by abiotic factors is typically attributed to a combination of decreased cell production and reduced mature cell size [17].

Plants undergo a series of physiological and morphological changes to cope with cold stress including increasing malondialdehyde (MDA) content, membrane permeability, proline accumulation, and altering antioxidant enzymes including superoxide dismutase (SOD), ascorbate peroxidase (APX), catalase (CAT), peroxidase (POD) and glutathione peroxidase (GPX) activities; such effects are generally achieved through significant transcriptional regulation [30]. However, an overreaction to any of these mechanisms may hinder seedling growth and biomass production, resulting in a trade-off between cold tolerance and overall yield. The influence of cold stress during early growth stages can significantly affect later growth phases. For example, prolonged cold stress through early planting has been found to induce cellular membrane damage and growth retardation in rice (*Oryza sativa* L.). via exposure to cold [31]. Plants promote flowering [19] and seed production [20] via the vernalization pathway, thus optimizing breeding [32]. Plants can enhance seed germination through cold stratification [33]. Experiences during one developmental stage can leave lasting impacts on later stages. Notably in arabidopsis (*A. thaliana*.), rosette vernalization has been shown to boost seed germination across various ecotypes [34] (Table 1). Cold stress at the reproductive stage leads to spikelet sterility [27], limits seed size [28], and ultimately leads to yield reduction (Table 1). For example, with each extra day of freezing at the critical temperature, plant mortality rates increased by 8.6%, 22.3%, 11.1%, and 9.4% for the wheat (*Triticum aestivum* L.) cultivars Jing411, Nongda211, Zhengmai366, and Yanzhan4110, respectively. Within the same cultivar, tillers demonstrated lower sensitivity to freezing duration (15–32% per day) when compared to younger leaves (25–35% per day) and older leaves (20–55% per day) [35]. Plant cold resistance can be reflected by measuring plant physiological indicators. This method is simple and easy to use. It is often used in cold resistance analysis and identification of plant resources, and is suitable for the screening and identification of various resources. It is also often used in combination with other biochemical markers. Traditional plant cold tolerance identification is still mainly performed by traditional evaluation methods using cold damage classification, and individual physiological and biochemical indicators for cold tolerance identification, which is simple and easy to implement and is usually used for plant resources, cold resistance analysis and identification. Therefore, it is necessary to identify and evaluate the cold tolerance of different plants, which can be reflected by measuring plant physiological indicators; however, some plant cold tolerance phenotypes are difficult to identify accurately, which directly impacts the effectiveness of the forward genetics approach in the discovery of cold tolerance genes.When temperatures drop below 0 °C and fall beneath the plant’s critical threshold, the formation of ice crystals within or on the surface of plant cells invariably leads to cellular damage and eventual plant death as ice crystals develop. Preventing ice crystal formation is regarded as a crucial mechanism of cold tolerance in plants. Cells may suffer mechanical damage or cellular dehydration resulting from cytoplasmic efflux due to an osmotic imbalance caused by the exclusion of solutes from recrystallized ice [36]. Downy birch (*Betula pubescens* Ehrh.) suffered no injury to any tissues under −70 °C cold stress [37]. Plants vary greatly in cold tolerance. It is of great significance to study the critical temperature of plants. Using the USDA Plant Hardiness Zone Map, the cold tolerance of plants can be preliminarily understood [38]. The impact of cold injury on plants depends on a variety of factors, including the intensity and duration of the cold injury, the stage of plant development at which cold injury occurs, and plant characteristics such as root structure, cuticle wax composition, and cell wall thickness [39]. Trichome coverage and 3D wax projections can be recognized as antifreezing strategies of plants, which increasing the thickness of the cuticle layer, stomatal density, and cuticular permeability [39]. Plants respond to cold stress through a series of physiological and morphological changes, such as ice-binding proteins and antifreezing proteins [36]. The ice-binding microalgal protein CmIBP1 effectively inhibits ice recrystallization both in vitro and in transgenic plants, as the intensity of IRI activity can be compared based on the size of the ice particles at a given temperature and duration [40]. CmIBP1 is secreted into apoplasts and improves freezing tolerance by inhibiting the growth of ice crystals in transgenic plants [40]. The reason for ice nucleation and freezing in plants is unknown. Studying cold perception in plants can provide the direction of ice diffusion from nucleation sites. Screening and cultivation of cold-tolerant germplasm resources based on strategies to prevent ice crystals could reveal ways to improve plant growth and defense.

### 2.1. Plant Cell Membrane

The normal functions and structures of cells are disrupted by the destruction of the plasma membrane and organelle membrane systems under cold stress. A study pointed out a direct relationship between the structure of plant cell membranes and plant cold tolerance [41]. At present, the mainstream view of plant perception of cold [42] is that cold damage occurs when the cell membrane transforms from a fluid state to a solid state, related to fatty acid changes (Figure 1). This hardening affects membrane proteins, leading to physiological and morphological changes with regulatory responses. Maintaining cell membrane integrity and fluidity is crucial for plants to cope with cold stress. Ice crystal formation can increase electrolyte leakage and causes lipid peroxidation [43]. Cold acclimation involves accumulation of unsaturated fatty acids and phospholipids [18,44], enhancing membrane integrity and cold tolerance (Table 1). Studies have shown that a specific haplotype of the *OsSRO1c* gene (*OsSRO1cHap1*) can significantly improve *O. sativa* cold tolerance at both seedling and heading stages. Further biochemical analysis shows that the OsSRO1c protein exhibits liquid–liquid separation characteristic properties in the nucleus and forms a condensate with OsDREB2B. These condensates can directly respond to cold stress and form small droplet structures through protein phase transformation [45]. The strong haplotype1 encoding the OsSRO1c protein d showed better mobility in condensed droplets formed by OsDREB2B and increased the transcriptional activity of OsDREB2B, thereby activating the first plant low-temperature sensors, including COLD1 [45] (Figure 1). This mechanism provides *O. sativa* with enhanced cold tolerance.

### 2.2. Chloroplast

Cold stress has a significant impact on plant growth and development, especially photosynthesis. Maintenance of normal chloroplast physiology is critical for plant growth and development. Under cold stress, the chloroplast structure is severely damaged due to excessive reactive oxygen species (ROS) accumulation (Figure 1). For example, cold stress induces overexpression of chloroplast Mpv17_PMP22 protein (MPD1) in *A. thaliana* which accelerates ROS generation and cold injury [46]. Chloroplasts are crucial targets for cold acclimation processes. Chlorophyll content is a key indicator of plant cold resistance. Chloroplast acclimation may be the limiting factor for cold adaptation [47]. Chloroplast damage affects photosynthesis and carbohydrate metabolism during cold acclimation [48]. Cold acclimatization in conifers involves seasonal changes in mesophyll cells. Chloroplast movement changes with season, suggesting that temperature drop dynamics depend on organelle movement [49]. To maintain PSII performance under cold stress, plants modify chloroplast structure. The seasonal movement of chloroplasts is mainly affected by low temperature stress. Under greenhouse conditions close to natural light and photoperiod, chloroplasts maintains their activity in the upper plate [49]. The timing of cytoplasmic cluster formation in the two *Picea* species studied coincides with the minimum seasonal level of the chlorophyll fluorescence parameter, characterizing the efficiency of the photosynthetic apparatus [49]. This period also corresponds to a decrease in grana development within the chloroplasts [49]. Triose phosphate and 3-phosphoglycerate export signals the chloroplast redox state, underlying cold photosynthesis [50]. Plant cold response depends on light quality, as cold exposure may induce PSII photoinhibition and oxidative damage [30]. Plants produce excess energy beyond their utilization capacity under stress [30]. This surplus energy leads to a decrease in the photosynthetic rate and electron transport capacity, resulting in photoinhibition. Although CA treatment can significantly mitigate the degree of PSII photoinhibition and oxidative damage in tobacco (*Nicotiana tabacum* L.) leaves under cold stress [30], seedlings can adapt to CA by regulating energy dissipation, thereby preventing excessive reduction in plastoquinone pools and subsequent PSII photoinhibition [30]. However, severe stress may progressively exacerbate PSII photoinhibition, if the excess excitation energy is not adequately dissipated through non-photochemical quenching pathways and other electron sinks in a timely manner. This further leads to increased ROS generation, thus damaging the photosynthetic mechanism [30]. The ability of many plant species to express robust phenotypes depends on light and photosynthetic activity during cold growth.

## 3. Mechanism of Cold Tolerance in Plants

Under cold stress, plants undergo various physiological and biochemical changes such as alterations in cell membrane permeability, osmotic regulators, peroxides, enzymes, and hormones. These changes are integrated into cold-induced signaling cascades to enhance cold tolerance. Plants perceive abiotic stress through key signal transduction pathways: abscisic acid (ABA) signal transduction pathway, Ca^2+^ signal transduction pathway and mitogen-activated protein kinase (MAPK/MPK) cascade pathway [51]. Cold stress modifies plant osmolytes, hormones, antioxidant systems, and transcription factors, enhancing ROS scavenging and upregulating cold-responsive genes [42]. ROS function as signaling molecules and regulate gene expression. This section explores plants’ cold stress tolerance mechanisms through phytohormones, plant growth regulators, ROS, protein kinases, and Ca^2+^.

### 3.1. Phytohormones and Plant Growth Regulators

Phytohormones play central regulatory roles in plant cold stress responses. Phytohormones are signal molecules that comprehensively respond to plant abiotic stress, and their signal transduction mechanism is related to the plant’s response to cold stress. Phytohormone signaling mediates abiotic stress responses through multiple mechanisms, including abscisic acid (ABA), gibberellin (GA), jasmonic acid (JA), salicylic acid (SA), brassinosteroids (BR), ethylene (ET), and melatonin (MET). As signaling molecules, they play a key role in activating the antioxidant enzyme system and regulating the expression of cold response genes, integrating external information into the cell interior, and influencing stress response pathways.

#### 3.1.1. Abscisic Acid (ABA)

The plant hormone ABA plays an important role in plant development and cold response, although its complex network signaling pathways remain unclear. ABA is a tolerance inducer, and its role in cold-signaling pathways can be divided into ABA-dependent pathways and ABA-independent pathways. The current consensus is that both ABA-dependent and ABA-independent pathways are involved in the cold-stress response of plants through the inducer of CBF expression (ICE)—C-repeat binding factor (CBF)—cold regulated (COR) pathway [52] (Figure 2). The study provides evidence that RCAR5/PYL11 exerts two distinct functions in seeds and leaves through Aba-dependent and Aba-independent pathways, respectively [53]. ABA-mediated overexpression of the *RCAR5* gene can inhibit both pre-germinative and post-germinative growth under cold stress conditions [53]. Regulation of cold stress in RCAR5 transgenic plants is controlled by OST1 and occurs independently of ABA [53]. Similarly, OST1-mediated phosphorylation of BTF3L positively regulates CBFs [54]. Phosphorylation of BTF3L Ser50 residue by OST1 is a necessary prerequisite for regulating antifreeze function [54]. ABA increases POD activity, SOD activity, and chlorophyll content, and reduces relative conductivity and MDA content to improve plant stability under cold stress [8]. Overexpression of *VaPYL9* increases the ability of tomato (*Solanum lycopersicum* L.) to scavenge ROS, reduces membrane lipid peroxidation and cell desiccation, thereby protecting *S. lycopersicum* from cold stress [55]. ABA-dependent cold-resistance pathways have been identified in a variety of plants. ABA usually accumulates rapidly under cold stress and regulates the expression of downstream stress response genes by integrating multiple stress signals.

#### 3.1.2. Jasmonic Acid (JA)

JA, as a key signal factor upstream of the ICE–CBF–COR transcriptional pathway, regulates plant cold resistance [56]. Jasmonate zim-domain 1(JAZ1) proteins, inhibitors of the JA signaling pathway (Figure 2), suppress cold stress responses in *A. thaliana* by inhibiting the transcriptional function of the transcription factor ICE1 [57]. Since JAZ protein contains a Jas domain that can interact with the receptor coronatine insensitive 1 (COI1) and a ZIM domain that can interact with Myelocytomatosis protein 2 (MYC2) of the bHLH family, it can physically interact with ICE1 of the bHLH family and repress its expression, thereby reducing cold tolerance [58]. When plants are subjected to cold stress, intracellular JA concentration increases, and JA-lle is transported to the nucleus through the JAZ1 protein, promoting the interaction of JAZ with the E3–ubiquitin ligase–SCF complex and the COI1 receptor [59,60]. This derepresses the ICE1 transcription factor through the JAZ protein, allowing the expression of downstream JA-responsive genes (Figure 2). Therefore, JA is a very important hormone in plants, which can improve plant resistance to cold stress and can interact with ABA metabolic pathways. Interestingly, in *Cucumis melo*, exogenous ABA and JA can improve cold tolerance, but their regulatory mechanisms are different. Exogenous ABA can improve cold tolerance of muskmelon by increasing the expression of *CmMYC2*, *CmPYL1* and *CmSOD1* genes and decreasing the expression of *CmPP2C3* [61] (Figure 2). JA may enhance cold tolerance of melon by decreasing the expression of *CmPP2C3*, *CmJAZ1* and *CmMYC2* [61] (Figure 2). MYC2 is a core transcription factor for JA signaling and a hub between ABA, SA, GAs, and IAA signaling [62]. MYC2 positively and negatively regulates multiple functions in the JA signaling pathway [62]. Physical interactions with other key regulatory proteins, formation of heterodimers with other transcription factors, and the ability to activate or repress gene expression in response to multiple signals can contribute to the diverse regulatory roles of MYC2 [62]. However, the antagonism and synergies between JA and other plant hormones in cold stress are unknown, and identifying key components integrating JA and other hormonal pathways is important.

#### 3.1.3. Gibberellin (GA)

GA is a vital plant growth regulator. GA, receptor GID1, and repressor DELLA shape the GA–GID1–DELLA module in the GA signaling cascade. DELLA, a key protein in GA signaling, is involved in CBF-regulated cold induction [63]. DELLA and JAZ interact through the Jas domain [64], and ICE1 is extracted from JAZ linked to DELLA (Figure 2). Also, the *CBF3* gene inhibits GA biosynthesis and promotes DELLA accumulation by activating GA2ox7 expression in *A. thaliana*, participating in CBF-regulated cold induction [65]. In cold temperatures, ICE1 is modified to perform the function of activating CBF3 transcription. CBF3 activates GA2ox7 and reduces bioactive GA levels, thereby promoting DELLA accumulation. Increasing DELLA releases more ICE1 to enhance the next round of CBF3 cold induction [66]. GA3 alone enhances cold resistance of flower buds but decreases it in branches, showing organ-specific effects. ABA/GA3 content better represents plant cold resistance than ABA alone [67]. Exploring plant hormones’ role in cold response, analyzing their regulatory network, and achieving hormone ratios/balances to regulate cold tolerance are crucial research directions.

#### 3.1.4. Auxin (IAA)

Under low temperature stress, exogenous IAA promotes root growth through *ARR1/12*, but cannot completely compensate for the inhibitory effect of low temperature on initial root growth in *A. thaliana* ecotype Columbia [68]. OsGH3-2 regulates the homeostasis of endogenous free IAA and ABA and has different effects on drought and cold tolerance of *Oryza sativa* subsp. japonica [69]. Transient cold stress at the tetrad stage of pollen development in *Brassica rapa* var. glabra Regel causes auxin-mediated starch-related energy metabolism imbalance that contributes to the decline in pollen germination rate and ultimately seed set [70]. SLR/IAA14 is a transcriptional repressor of IAA signaling and plays a crucial role in microRNA integration into IAA and cold response.

#### 3.1.5. Salicylic Acid (SA)

SA combats abiotic stresses in plants and mitigates cold-induced changes. A new model [71] shows that SA activates ROS signal integration into Ca^2+^/CPK-dependent ABA signal transduction branches rather than stomatal opening factor 1 (OST1) signal transduction branches (Figure 2). Exogenous SA enhances cold response gene expression, PSII, Fv/Fm, and Pn, and reduces MDA. SA regulates cold-induced changes via ABA-dependent or independent pathways, Ca^2+^ signaling pathways, MAPKs pathways, and ROS pathways [72]. SA can improve freeze tolerance of *Triticum aestivum* leaves by affecting apoplastic proteins [73]. Exogenous SA can lead to increased ice nucleation activity at low temperatures (15/10 and 10/5 °C) [73]. SA may participate in cold resistance by regulating the activities of winter wheat (*T. aestivum* cv. Dogu-88) axoplast proteins and antioxidant enzymes, reducing the activities of CAT and POX, and increasing the activities of polyphenol oxidase [74]. SA-treated cells can maintain Ca^2+^ homeostasis under cold or heat stress and improve the tolerance of young grapes (*Vitis vinifera* L.) [75]. The alleviating effect of SA on chilling injury of peaches (*Prunus persica* (L.) Batch. cv. Beijing 24) during cold storage may be attributed to its ability to induce antioxidant systems and heat shock proteins. Cold stress activates the innate immune response through the SA-dependent pathway. Effects of 10 h cold-temperature treatment in *A. thaliana* were similar to those caused by pathogen infection, including increased expression of the SA pathway marker genes PR2 and PR5, as well as increased expression of genes that play an active role in defense against (hemi)-biotrophs [76]. After cold stress treatment, transcripts encodings some SA biosynthetic enzymes (but not ICS1/SID2) were more abundant, while transcripts encoding components involved in SA modification were less abundant [76]. However, it is unclear whether long-term cold stress-induced disease resistance is related to early short-term cold stress-induced responses.

#### 3.1.6. Brassinosteroids (BRs)

BRs are widely used in agriculture, which impacts plant response to cold stress [77]. Brassinosteroid (BR) receptor brassinazole-resistant 1 (BZR1) regulates *CBF1* and *CBF2* gene expression [78], enhancing cold resistance without affecting growth (Figure 2). BRs actively control cold stress response by promoting expression and cold tolerance of CBF1 and CBF-regulated gene *COR47* in *A. thaliana* [79]. PIF4 protein accumulation and transcriptional activity are regulated by light and temperature, linking the exogenous environment with endogenous BR levels [80]. An amount of 1 μM BRs can change the cold acclimation process by stimulating photosynthesis and carbohydrate metabolism, thereby improving cold tolerance of winter rye (*Secale cereale* L.)[81]. The crosstalk of H_2_O_2_ and NO is involved in the BR-induced cold tolerance of *Medicago truncatula* [82]. PSII efficiency induced by BRs under stressful conditions may depend on NO production rather than H_2_O_2_ production [82]. Endogenous BRs do not travel far, thus regulation of BR signal transduction at the specific tissue or organ level is crucial for plants to respond to cold stress.

#### 3.1.7. Ethylene (ET)

ET is involved in regulating a series of biological processes. The ET signal pathway may inhibit the CBF transcriptional cascade in soybeans (*Glycine max* (L.) Merr.) through the action of ethylene-insensitive protein 3 (EIN3) [83] (Figure 2). Similarly, during cold storage of post-harvest loquat fruit, ethylene signal transducers were involved in the lignification process of cold injury through different regulatory sites to alleviate cold injury [84]. The expression of *LeCBF1* is regulated by exogenous ethylene and 1-methylcyclopropene and is not expressed without cold induction in postharvest *S*. *lycopersicum* [85]. MtSKL1 is required for detecting changes in ET level in *M. truncatula* plants for the full development of the cold acclimation response by suppressing expression of *MtEIN3* and *MtEIN3/EIL1*, which in turn downregulates expression of *MtERFs*, leading to the enhanced tolerance of *M. truncatula* to freezing by upregulating *MtCBFs* and *MtCAS15* [86]. The MdERF1B–MdCIbHLH1 regulatory module plays a role in ET-mediated cold stress responses in *Malus pumila* [87] (Figure 2). Cold stress rapidly induces ethylene production and upregulates MdERF1B expression. Additionally, MdERF1B interacts with MdCIbHLH1 to activate MdERF1B-mediated cold tolerance and ethylene biosynthesis [87].

#### 3.1.8. Cytokinin (CTK)

Under transition zone climate conditions, choosing to use seedlings with higher ABA values and CTK values can improve the winter survival and spring vegetation recovery of *Cynodon* spp. [88]. *Verticillium dahliae* Aspf2-like protein (VDAL) promoted potato (*Solanum tuberosum* L.) growth, particularly at low temperatures [89]. Time-course transcriptomic analysis and endogenous phytohormone detection revealed that CTK may play an important role in response to VDAL-promoted growth [89]. Phytohormone IAA and CTK signaling directly or indirectly regulate gravitropism response and root development under low-temperature stress [25]. Furthermore, the protective mechanisms against DNA damage in root stem cells induced by low-temperature stress involves the crosstalk of IAA and CTK. AHK2 and AHK3 and the cold-inducible ARRs play a negative regulatory role in cold stress signaling by inhibiting ABA responses, independent of the cold acclimation pathway in *A. thaliana* [90] (Figure 2). Moreover, transient expression of a subset of ARR genes, including ARR5, ARR6, ARR7, and ARR15, which are downstream targets of EIN3, may reveal an inherent connection between cold stress and ET [91] (Figure 2). ARRs are important nodes that integrate ET and CTK signals to control plant responses to environmental stress [91].

#### 3.1.9. Melatonin (MET)

Exogenous MET can improve plant growth and tolerance to cold stress by regulating antioxidant enzymes. Pretreatment with 1 μM melatonin restores rhythmic accumulation of hydrogen peroxide, SOD, and CAT in hull-less barley seedlings under cold stress [92]. Melatonin scavenges reactive oxygen species (ROS) by accelerating the AsA-GSH cycle, balancing photosynthetic degradation, and inhibiting ROS production, enhancing cold tolerance in cucumber seedlings [93]. Exogenous MET application can increase the expression of *CBFs*, *COR15a*, *ZAT10* and *ZAT12*, thereby improving the growth and cold tolerance of *A. Thaliana* [94] (Figure 2). MET alleviates cold stress by upregulating the expression of *CsZat12* and regulating the metabolism of polyamines and ABA [95]. MET might activate MAPK through H_2_O_2_ or Ca^2+^ dependent pathways in *A. thaliana* [96] (Figure 2). Activated MAPK can phosphorylate and activate SOG1 and inhibitory MYB to alleviate DNA damage under abiotic stress [96]. Injecting MET into the mother plant during the grain filling period can promote the germination of offspring seeds by accelerating starch degradation, and improve the cold tolerance of seedlings by activating antioxidant enzymes and improving photosynthetic electron transfer efficiency [97]. MET is unstable, and further studies are needed to explore its interactions with other penetrants and hormones under cold stress, as well as MT signaling between organelles.

### 3.2. Signalling Compounds

#### 3.2.1. Reactive Oxygen Species (ROS)

ROS signaling under cold stress generates excess ROS, leading to oxidative damage and cold stress response in plant cells. Boosting antioxidant enzymes to clear excess ROS is a key strategy for plants to handle cold stress. PCA analysis of mangrove species showed high antioxidant enzyme activity is crucial for cold tolerance [98]. tAPX triggers expression of COR15A, PAL1, and CHS proteins in response to cold stress [99]. Long-term exposure to cold stress showed that sAPX was not relevant and showed a strong dependence on tAPX [98]. Thylakoid protection mediated by tAPX acts as an initiation center that stores initiation information over time [99]. tAPX-mediated thylakoid protection serves as a priming hub, which stores information on priming over time [99]. Compared with plants induced by short-term cold stress, long-term cold stress induced stronger induction of non-chloroplast-specific ROS-regulated genes such as CHS and PAL1 (and COR15A), which support salicylic acid, lignin, flavonoids, and floral biosynthesis of various secondary stress protective mediators such as anthocyanins [99]. Heme-associated protein AtHAP5A enhances cold tolerance and suppresses ROS accumulation by binding *AtXTH21* [100]. Genetic evidence indicated that *AtHAP5A* acts upstream of *AtXTH21* in freezing stress response in *A. thaliana* [100]. These results revealed that AtHAP5A modulates freezing stress resistance through interaction with the CCAAT motif of *AtXTH21* in *A. thaliana* [100]. Overexpressing *AtHAP5A* and *AtXTH21* could alleviate 4 °C stress-induced ROS accumulation and related oxidative damage in *A. thaliana*, while *AtHAP5A* and *AtXTH21* mutants had the opposite effect [100]. Overexpression of the ROS signal response gene *AtZAT12* downregulates *AtCBF1/2/3* genes under cold stress [101,102] (Figure 2). ZAT12 downregulated the expression of the *CBF* genes indicating a role for ZAT12 in a negative regulatory circuit that dampens expression of the CBF cold response pathway [101]. The role of ZAT12 regulation may help plants cope with oxidative stress. In this regard, it is of great interest that ZAT12 expression resulted in downregulation of transcripts encoding a putative l-ascorbate oxidase [101]. Cold triggers ROS accumulation in plant cells, reducing protein content, enzyme activities, and expression of cold resistance genes through biochemical reactions.

#### 3.2.2. Protein Kinases

Histidine protein kinase (HPK) may act as a cold sensor. Upon receiving external stimuli, histidine residues are phosphorylated and passed to response regulatory proteins (RR), which receive phosphate groups and transmit signals. The second messenger regulates kinases like Ca^2+^-dependent protein kinases (CDPKs) and receptor-like protein kinases (RLKs) (Figure 1), while protein phosphatase (PP) regulates proteins and transmits signals to downstream cold-signaling pathways. OsCPK24, a cold-reactive kinase on the cell membrane, enhances cold tolerance by stimulating thiol transferase activity of OsGrx10 in response to proline and Ca^2+^, regulating glutathione levels to combat cold stress [103].

The mitogen-activated protein kinase (MAPK/MPK) cascade is an important way to sense cold signals and regulate gene expression [104]. H_2_O_2_ is a signal molecule to excite it [105]. The AtCRLK1-AtMEKK1/2-AtMPK4/6 cascade improves cold tolerance by antagonizing the AtMPK3/6 pathway [106,107]. MtCTLK1 or MfCTLK1 affect cold resistance through the CBF transcriptional cascade, antioxidant defense, and proline accumulation [108].

#### 3.2.3. Ca^2+^

Cold stress increases intracellular second messenger Ca^2+^, which is transmitted to downstream cold-signaling pathways, leading to transcriptional remodeling in plants. Calmodulin (CaM) and CaM-like proteins (CML), Calcineurin B-like (CBL), and calcium-dependent protein kinases (CPKs), as three calcium signaling receptor proteins, are mainly involved in the perception and transmission of cold signals in plants [109].

Calmodulin-binding transcription activator (CAMTA), MYB, WRKY, NAC, basic leucine zipper (bZIP), and MADS-box are calmodulin-binding proteins (CaMBPs) (Figure 3). Cyclic nucleotide-gated channels (CNGCs), nonselective cation channels in plant cell membranes, are key components of plant temperature perception [110,111]. Previous analyses of plant CNGC family protein structures found that they all contain CaM-binding sites and that there is a physical link or functional interaction between CaM and CNGC [112,113].

CBL is a calcium-binding protein involved in plant signaling. AtCBL1-AtCIPK3 regulate cold stress tolerance in *A. thaliana* by transmitting signals to cold resistance-related transcription factors [114]. CBL2, CBL3, CIPK9, and CIPK27 negatively regulate the ABA signal pathway in *A. thaliana* [115]. OsCBL2 is the only CBL up-regulated in the *O. sativa* aleurone layer induced by GA [116]. AtCBL3 interacts with AtMTN1 to regulate ethylene biosynthesis and polyamines during plant growth [117]. CBL6/CBL8-CIPK14 is uniquely responsive to cold in citrus [118]. Interestingly, HvCBL4 maintains circadian rhythm at different temperatures, aiding in studying the effect of the biological clock on grain cold resistance [119].

In contrast to CaM and CBL, Ca^2+^ signals can be directly converted into a single CPK protein phosphorylation event [120]. In 1982, CPK was first identified in peas (*Pisum sativum* L.) [121]. Some CPKs are Ca^2+^-dependent, while others are Ca^2+^-independent (such as AtCPK7 and AtCPK32) [122]. CPK interacts with plant hormones during cold stress [123,124]. CPK11 is induced by H_2_O_2_ and regulates antioxidant enzymes through the ABA signal pathway [125,126]. Overexpression of *VaCPK20*, a calcium-dependent protein kinase gene, enhances drought and cold tolerance by regulating proteins [127]. In *O. sativa*, OsCPK24 is a cold-reactive kinase that enhances cold tolerance by regulating glutathione levels [103].

## 4. Downstream Mechanisms and Regulation for Cold Signaling in Plants

Cold tolerance in plants is a quantitative trait controlled by multiple genes, which are expressed synergistically under the regulation of transcription factors, and multiple signaling pathways jointly regulate plant responses to cold stress (Figure 2) [128]. The transcriptional regulation of plant cold-tolerance genes can be divided into two types: CBF-dependent and CBF-independent [129]. The CBF-regulated signaling pathway is currently the most studied transcription factor regulating cold-induced gene changes.

### 4.1. Transcription Factors in Plant Cold Signaling

Transcription factors have always been a hot spot in plant cold stress research, and they play an important regulatory role in signal transduction and gene expression under cold stress [130]. They bind to *cis*-acting elements upstream of stress-responsive genes to activate or repress gene expression [131]. For example, MYB, WRKY, NAC, and bZIP are the transcription factors which belong to calmodulin-binding proteins, and play an important role in plant cold induction (Figure 3) [132]. Hypothermia receptors mostly trigger calcium ions. It is important to create a suitable expression pattern for positive regulators of cold resistance without negatively affecting favorable agronomic traits. However, studies on these transcription factors have mainly focused on their functional mechanism, and little is known regarding the effects of phytohormones on the activity of these proteins.

#### 4.1.1. AP2/ERF

Aptala2/ethylene response factor (AP2/ERF) is the most studied transcription factor involved in the mechanism of cold response [133]. C-repeat binding factor/dehydration response element binding factor 1 (CBF/DREB1) belongs to its subfamily, and there are many studies on its regulation under cold stress [51]. Most studies on cold signaling in plants still revolve around the ICE–CBF–COR transcriptional pathway [134]. Through this pathway, CBF transcription factors activate *COR* gene expression by binding to CRT/DRE elements in the *COR* promoter, enhancing cold tolerance [135]. ABA increases CBF transcriptional level, possibly through binding to CRT/DRE elements [136], which activates COR genes to improve cold tolerance. There are three main *CBF* genes: *CBF1*, *CBF2*, and *CBF3*. AtCBF1/3 transcription factors improve cold tolerance by binding to CRT/DRE elements of *COR* genes [137] (Table 2). However, *AtCBF2* negatively regulates *CBF1* and *CBF3*, reducing cold resistance [138]. The mechanism by which CBF mediates cold stress and its regulatory mechanism need to be explored. Overexpression of *CBF* enhances cold resistance but may affect plant growth. CBF transcription factors are also regulated by hormones (Figure 2).

#### 4.1.2. MYB

MYB transcription factors play a two-way role in plant stress response (Table 2). On one hand, transcription factors *A. thaliana* AtMYB15 [139,140], *Musa nana* MpMYBS3 [141], and *O. sativa* OsMYBS3 [142] are inhibitors of cold signals by inhibiting the expression of *CBF* genes under cold stress. On the other hand, overexpression of *MYB96* in plants can result in strong cold tolerance [143]. Interestingly, while the MYB15 transcription factor negatively regulates the *CBF* genes in *A. thaliana* [139] and pepper *(Capsicum annuum* L.) [144], SlMYB15 transcription factor positively regulates the *CBF* genes, which shows MYB15 in different plants has different responses to cold stress [145]. The OsMYB30 transcription factor regulates starch decomposition and cold tolerance by negatively regulating the expression of the α-amylase 1a (*OsAMY1a*) gene in *O. sativa* [146]. OsMYB30-OsTPP1 is a sugar signaling pathway that regulates seed germination in response to low temperature. Expression of *OsMYB30* and *OsTPP1* was induced by low temperatures during seed germination [146]. OsMYB30 binds to the promoter region of *OsTPP1* to activate its expression [146]. Overaccumulation of trehalose was found in both OsMYB30- and OsTPP1-overexpressing lines, resulting in inhibition of *OsAMY1a* during seed germination [146] (Table 2). The CaMYB306 transcription factor inhibits the positive cold resistance regulator CaCIPK13 in *C. annuum* [147]. It also suppresses chlorophyll and anthocyanin contents and regulates ROS signaling.

#### 4.1.3. WRKY

In 1994, Ishiguro and Nakamura found the first WRKY transcription factor SPF1 in sweet potato (*Ipomoea batatas* (L.) Poir.) [148]. WRKY mainly responds to a variety of induced stimuli by specifically recognizing W-Box and initiating the expression of regulatory genes, thus playing a regulatory role in plants [149]. The transcription factor AtWRKY34 (Table 2), specifically expressed in pollen, may participate in the cascade of CBF signals in mature pollen and negatively mediate the cold sensitivity of mature *A. thaliana* pollen [150]. Similarly, WRKY22 transcription factor is essential for cold stress adaptation in *A. thaliana* buds (Table 2). It alters SA-mediated wounding and osmotic stress responses in *A. thaliana* [151]. It suppresses the transcription of *WRKY53* and *WRKY70*. *WRKY53* gene expression induces SA expression [152,153]. In *A. thaliana*, *AtWRKY70*, downstream of SA receptor NPR1, is regulated by SA and JA [154]. Glutathione regulates MPK3 expression through *WRKY40* [155]. Bermuda grass (*Cynodon dactylon* (L.) Pers.) CdWRKY2 and *S*. *lycopersicum* SlWRKY50 increase cold resistance via a CBF-dependent pathway [156,157]. In cucumber (*Cucumis sativus* Linn.), CsWRKY46 [158] enhances cold resistance via an ABA-dependent pathway (Table 2). Future research should focus on mining more target genes and exploring the interaction between transcription factors, WRKY, and other proteins to understand the regulatory mechanism.

#### 4.1.4. bZIP

ACGT element-specific plant DNA-binding proteins belong to the bZip domain [159]. ABERs are bZIP transcription factors. AREB/ABF transcription factors regulate ABA-related gene expression and cold stress. Proteins like ABF1, ABF2/AREB1, and ABI5 are involved in ABA signal transduction under stress [160]. Chorispora bungeana (*Chorispora bungeana* Fisch. et Mey.) CbABF1 confers stress tolerance in *N. tabacum* by reducing ROS and enhancing antioxidant enzymes [161]. *Actinidia eriantha* Benth. *AcePosF21* [162] reduces ROS damage by mediating *AceGGP3* expression (Table 2). The transcription factor ZmbZIP68 [163] inhibits the CBF-signaling pathway in *Z. mays* (Table 2). In *A. thaliana*, tea (*Camellia sinensis* (L.) O. Kuntze) CsbZIP18 [164] negatively regulates cold tolerance through an ABA-dependent but not CBF-dependent pathway (Table 2). LIP19/OsOBF1 [165] heterodimer formation improves cold tolerance in *O. sativa* (Table 2). GmbZIP44, GmbZIP62, and GmbZIP78 play a negative role in ABA signaling by up-regulating ABI1 and ABI2 proteins, inducing genes like *ERF5KIN1*, *COR15*, and *COR78* for salt and cold tolerance in transgenic plants [166].

Elongated hypocotyl5 (HY5) is a key bZIP transcription factor involved in crosstalk between light and cold response pathways, integrating ABA and ROS signaling to attenuate photo inhibition (Figure 2). The SlHY5 transcription factor binds to the promoter of *ABI5*, triggering respiratory burst oxidase homolog1 (RBOH1)-dependent H_2_O_2_ production in the apoplast [167] (Figure 2). The CRYPTOCHROME2 (CRY2)-COP1-HY5-BBX7/8 module regulates blue light-dependent cold acclimation in *A. thaliana* [168]. Additionally, BBX7 and BBX8 act as HY5 targets, positively regulating freezing tolerance by modulating cold-responsive gene expression, primarily in CBF-independent pathways [168].

#### 4.1.5. NAC

NAC (NAM, ATAF1/2, CUC2) is one of the largest families of transcription factors endemic to plants [169]. *A. thaliana* NAC transcription factor JUNGBRUNNEN1 (AtJUB1) inhibits GA3ox1 and DWF4, and reduces PIF4 expression [170]. GA mutant *cpk28* alters expression of NAC transcriptional regulators and GA3ox1 [171,172]. NAC transcription factors participate in plant growth, development, and cold stress response through hormone signal cross-pathways. MaNAC25 and MaNAC28 form a positive feedback loop, negatively regulating cold tolerance in banana (*M. nana Lour.*) fruit by up-regulating phospholipid degradation gene expression (Table 2) [172]. The CaNAC1 transcription factor can also negatively regulate cold stress in plants through phospholipid degradation (Table 2) [173]. *C*. *annuum* CaNAC064 and siberian crabapple (*Malus baccata* Borkh) MbNAC25 transcription factors regulate plant cold tolerance by enhancing ROS scavenging capacity in *A. thaliana* (Table 2) [174,175]. HuNAC20 and HuNAC25 transcription factors have cold tolerance in transgenic *A. thaliana* [176].

#### 4.1.6. bHLH

bHLH is a conserved basic/helix–loop–helix domain with a basic amino acid region and a helix–loop–helix region [177]. ICE1 and ICE2 (e.g., *Hevea brasiliensis* (Willd. ex A. Juss.) Müll. Arg. HbICE2) bHLH-type transcription factors are positive regulatory factors of CBF [178]. Snow lotus (*Saussurea involucrata* (Kar. & Kir.) Sch. Bip.) SiICE1 (Table 2) binds to MYC in the *CBF3* promoter, promoting *CBF3* expression and cold tolerance in *S*. *lycopersicum* [179,180]. MeJA positively regulates *HbICE2* overexpression in *A. thaliana*, enhancing cold tolerance [178]. Overexpression of ZjICE2 improves chlorophyll content, photosynthetic efficiency, and cold resistance injapanese lawngrass (*Zoysia japonica* Steud.) (Table 2) [181]. *IbbHLH116* functions like *ICE1*, conferring cold tolerance to *I. batatas* (Table 2) [182]. The MdbHLH4 transcription factor represses *MdCBF1/3* expression, promotes *MdCAX3L-2* expression, and negatively regulates cold tolerance in *M. pumila* domestica (Table 2) [183].

Phytochrome-interacting factor (PIF) belongs to the 15th subfamily of the bHLH transcription factor family. PIF3, the first plant pigment interaction factor, was discovered in 1998 [184]. Under cold stress, red/far red light induces *S. lycopersicum* PIF4 accumulation via phytochrome A (phyA). SlPIF4 binds to the *S. lycopersicum* CBF promoter to promote expression and positively regulate cold tolerance [185]. Additionally, SlPIF4 activates *SlDELLA4* expression in the GA signaling pathway, with SlGAI4 regulating tomato cold tolerance (Figure 2). When SlGAI4 accumulates, it inhibits SlPIF4, forming a negative feedback loop, and PIF4 transcription factor, forming a negative feedback regulation loop (Table 2) [185]. Further research is needed on PIF’s role in balancing stress and growth through hormonal pathways and adapting to changing environments.

MYC2 is also a bHLH transcription factor and the main regulator of the JA signal pathway. The PtrMYC2 (Table 2) transcription factor integrates JA signals by directly regulating the accumulation of PtrBADH-1 and GB in trifoliate orange (*Poncirus trifoliata* (L.) Raf.), thus regulating cold induction [186]. The SIMYC2 (Table 2) transcription factor may be involved in the cold tolerance induced by MeJA, possibly by improving the antioxidant enzyme system and increasing the levels of proline and lycopene in fruits [187].

**Table 2 cells-14-00110-t002:** Identified gene resources related to cold response in plants.

Families	Genes	Species	Cold Stress Regulation	References
AP2/ERF	*AtCBF1/DREB1B*	*A. thaliana* (L.) Heynh.	Positive	[137]
	*AtCBF2/DREB1C*	*A. thaliana* (L.) Heynh.	Negative	[138]
	*AtCBF3/DREB1A*	*A. thaliana* (L.) Heynh.	Positive	[137]
MYB	*AtMYB15*	*A. thaliana* (L.) Heynh.	Negative	[139,140]
	*SlMYB15*	*S. lycopersicum* L.	Positive	[145]
	*AtMYB96*	*A. thaliana* (L.) Heynh.	Positive	[188]
	*MpMYBS3*	*M. nana* Lour.	Positive	[141]
	*OsMYB30*	*O. sativa* L.	Negative	[189]
	*CaMYB306*	*C. annuum* L.	Negative	[147]
WRKY	*AtWRKY34*	*A. thaliana* (L.) Heynh.	Negative	[150]
	*AtWRKY22*	*A. thaliana* (L.) Heynh.	Positive	[151]
	*AtWRKY40*	*A. thaliana* (L.) Heynh.	Negative	[155]
	*SlWRKY50*	*S. lycopersicum* L.	Positive	[156]
	*CdWRKY2*	*C. dactylon* (L.) Pers.	Positive	[157]
	*CsWRKY46*	*C. sativus* L.	Positive	[158]
bZIP	*CbABF1*	*C. bungeana Fisch. et Mey.*	Positive	[161]
	*AcePosF21*	*A. eriantha* Benth.	Positive	[162]
	*ZmbZIP68*	*Z. mays* L.	Negative	[163]
	*CsbZIP18*	*C. sinensis* (L.) O. Kuntze	Negative	[164]
	*OsbZIP38/LIP19*	*O. sativa* L.	Positive	[165]
	*SlHY5*	*S. lycopersicum* L.	Positive	[145,167]
NAC	*AtJUB1*	*A. thaliana* (L.) Heynh.	Negative	[170]
	*MaNAC25*	*M. nana* Lour.	Negative	[172]
	*MaNAC28*	*M. nana* Lour.	Negative	[172]
	*CaNAC1*	*C. annuum* L.	Negative	[173]
	*CaNAC064*	*C. annuum* L.	Positive	[174]
	*MbNAC25*	*M. baccata* Borkh.	Positive	[175]
bHLH	*HbICE2*	*H. brasiliensis* (Willd. ex A. Juss.) Müll. Arg.	Positive	[178]
	*SiICE1*	*S. involucrata* (Kar. & Kir.) Sch. Bip.	Positive	[180]
	*ZjICE2*	*Z. japonica* Steud.	Positive	[181]
	*IbbHLH116*	*I. batatas* (L.) Poir.	Positive	[182]
	*MdbHLH4*	*M. pumila* Mill.	Negative	[183]
	*SlPIF4*	*S. lycopersicum* L.	Positive	[185]
	*PtrMYC2*	*P. trifoliata* (L.) Raf.	Positive	[186]
	*SlMYC2*	*S. lycopersicum* L.	Positive	[187]

### 4.2. Post-Translational Modification in Plant Cold Signaling

#### 4.2.1. Sumoylation (SUMO)

Under cold stress, the ubiquitin-like modifier E3 ubiquitin ligase SIZ1 positively regulates the ICE1 transcription factor via sumoylation (SUMO) [190,191] (Figure 2). High HOS1 expression negatively regulates ICE1 through ubiquitin modification, affecting CBF expression [192]. BIN2 phosphorylates ICE1, promoting its interaction with E3 ubiquitin ligase HOS1, leading to ICE1 degradation and reduced CBF expression [193].

#### 4.2.2. Protein Phosphorylation

Protein phosphorylation is a post-translational modification in plant cold signaling. CBL9-CIPK3 phosphorylates ABI1 to form CBL9-CIPK3-ABR1 in *A. thaliana* [194]. Calcium-dependent protein kinase 21 phosphorylates 14-3-3 proteins in response to ABA signaling and salt stress in rice [195]. Cold stress activates OST1, which regulates CBF stability through the OST1-BTF3 complex [54] (Figure 2). OST1 phosphorylates BTF3 and BTF3L to promote CBF stability under cold stress (Figure 2). *MaNAC1* is a target of MaICE1 and interacts with MaCBF1 to enhance cold stress response. MaICE1 phosphorylation and cold stress improve *MaNAC1* binding, leading to cold tolerance in *M. nana* through the ICE-CBF-COR pathway [196].

#### 4.2.3. Mitogen-Activated Protein Kinase (MAPK/MPK) Cascade

In *A. thaliana*, the MAPKKK protein AtANP1 triggers the phosphorylation cascade with AtMPK3 under cold stress [197]. The AtCRLK1–AtMEKK1/2–AtMPK4/6 cascade enhances cold tolerance by counteracting the AtMPK3/6 pathway [106]. AtMPK3/6 phosphorylates AtMYB15 and reduces the binding affinity of the AtCBF3 gene, resulting in enhanced transcriptional inhibition of CBF3 by MYB15, negatively regulating cold tolerance [139]. Glutathione regulates MPK3 expression through WRKY40 [155]. MPK3/6, negatively regulated by MKP1 [198], plays a key regulatory role during cold acclimation and may enhance freezing resistance and plant survival [199]. It phosphorylates CAMTA3, contributing to its degradation and inhibition of downstream target gene expression [200].

### 4.3. Multi-Omics Analysis Facilitates the Identification of Cold Resistance Genes

Advances in plant multi-omics research have promoted the discovery of plant cold tolerance molecular mechanisms. Using transcriptomes, *S*. *lycopersicum* SlWRKYs were induced by cold stress with CRT/DRE elements in SlWRKY2 and SlWRKY50 promoters, suggesting their role in CBF-mediated cold response [156]. One-month-old *A. thaliana* was exposed for 1 week to 4 °C in short-day conditions under white (100 and 20 μmol m^−2^s^−1^), blue, or red (20 μmol m^−2^s^−1^) light conditions [201]. Proteomic data showed distinct differences between red and blue light-treated plants under cold stress, with blue light activating cold-related proteins and red light upregulating chloroplast proteins [201]. Comparative proteomic analysis of chloroplast proteins in sugar beet *(Beta vulgaris* L.) provided insights into cold effects. It copes with cold stress by transporting photosynthetic proteins, forming starch granules, and scavenging ROS [202]. Advances in plant multi-omics, such as genomics, transcriptomics, proteomics, microbial functional genomics, epigenetics, and metabolomics, have accelerated the discovery of the molecular mechanisms behind plant cold tolerance [203].

## 5. Conclusions and Future Perspectives

In conclusion, this review discusses recent research into the ways some plants cope with cold stress, and regulatory mechanisms of cold tolerance in plants. Cold stress can quickly induce the expression of many transcription factors, thus activating a large number of downstream cold response gene expression transcription factors that play a key role in regulating gene expression (Figure 3). Plant hormones as signaling molecules are involved in the regulation of plant cold response, and the coordination of hormone and cold-signaling pathways can better deal with cold stress (Figure 4). At present, these studies focus on the function of a single gene, most of which are CBF-dependent pathways. However, the cold response transcriptional group regulated by CBFs accounts for only about 12% [204]. Therefore, further screening of various cold-related genes and the interaction mechanism between them is the key to exploring the mechanism of signal transduction and regulation in the future.

At present, the mechanisms of cold response regulation and control networks are not fully elaborated, and cannot be implemented in production practices. The phenotypes of different growth stages may be different. It is necessary to explore the evaluation system to use future genetic methods to mine excellent cold tolerance genes, especially cold response genes specific to a single species. Tissue-specific expression or induced expression is mainly used to reduce the effect of the constitutive expression on the growth of the plant sensors. How plants perceive cold is still unclear. Finding the upstream temperature signal sensor of the plant cold-signaling pathway is a fundamental problem in future research because these receptors are very likely to play a switching function for the whole cold-signaling pathway. Interestingly, there is a close relationship between drought stress and cold stress. MYB transcription factor gene *ApMYB77* [205] and *MbMYB4* [206] confer both freezing and drought tolerance. *MdBES1* was a positive regulator for cold tolerance and disease resistance in *M. baccata* Borkh, but a negative regulator for drought tolerance [207]. Dong et al. (2019) discovered the regulation of leaf-derived jasmonic acid as a long-distance transport signal to regulate water uptake by cotton (*Gossypium hirsutum* L.) roots and invented a liquid fertilizer that promotes jasmonic acid synthesis under partial root-zone irrigation to cope with drought [208]. But there are still few corresponding hormone fertilizers for cold stress, so further research is necessary for the construction and mining of the cold stress network. This will be helpful to develop cultivation and management methods to enhance plant cold tolerance (such as the application of exogenous substances) to improve production.

An increase in extreme weather will amplify the impact of cold stress on agricultural production. A future challenge for improving crop cold tolerance is to combine knowledge gained from model systems with multi-omic and genetic data for new crop varieties and crop performance test systems. In addition, efforts should be made to identify natural cultivars of unknown stress-resistant resources and understand their underlying mechanisms. As more powerful resources are discovered and identified, these efforts must be effectively integrated into plant breeding to achieve sustainable global food security.

## Figures and Tables

**Figure 1 cells-14-00110-f001:**
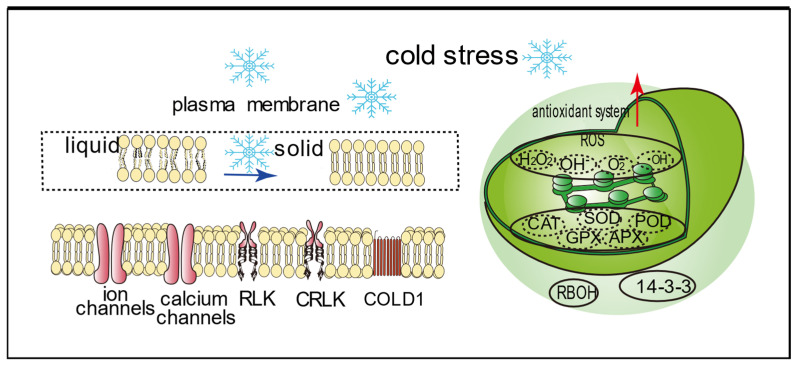
Cold stress impact on plants. When plants are subjected to cold stress, their cell membrane system and chloroplast antioxidant system are disrupted, and the cell membrane changes from a fluid-liquid-crystal state to a solidified gel state, with accelerated synthesis of ROS. Ion channels, calcium channels, RLK, CRLK, and COLD1 act as receptors. Abbreviations: RLK, receptor-like protein kinases; CRLK, calcium receptor-like protein kinases; COLD1, chilling tolerance divergence 1; ROS, reactive oxygen species; SOD, superoxide dismutase; CAT, catalase; POD, peroxidase; GPX, glutathione peroxidase; APX, ascorbate peroxidase; RBOH, respiratory burst oxidase homologue; H_2_O_2_, hydrogen peroxide; O^2−^, superoxide anion; ·OH, hydroxyl radical; OH^−^, hydroxyl; 14-3-3, 14-3-3 protein.

**Figure 2 cells-14-00110-f002:**
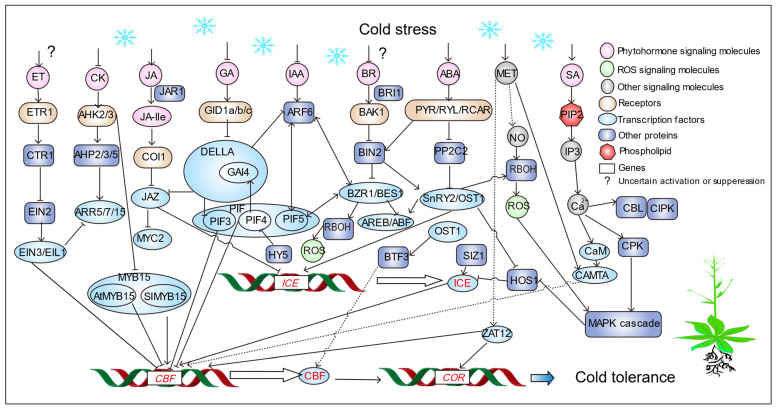
The mechanism of cold stress tolerance in plants. Arrows and lines ending in bars indicate activation and suppression processes, respectively. Solid lines represent direct interactions and dotted lines indicate indirect interactions. See the text for additional details. Abbreviations: ET, ethylene; CK, cytokinins; JA, jasmonate; GA, gibberellins; IAA, indole acetic acid; BRs, brassinosteroids; ABA, abscisic acid; SA, salicylic acid; MET, melatonin; ETR1, ethylene response 1; CTR1, constitutive triple response 1; EIN, ethylene-insensitive; AHK, *A. thaliana* histidine kinase; AHP, histidine phosphotransfer protein; ARR, *A. thaliana*. response regulator; JAR1, jasmonate resistant 1; COI1, coronatine insensitive 1; JAZ, jasmonate ZIM-domain protein; MYC2, myelocytomatosis proteins 2; PIF, phytochrome-interacting factor; GID, gibberellin-insensitive dwarf 1; GAI, gibberellic acid insensitive; ARF, auxin response factor; BRI1, brassinazole-resistant 1; BAK1, BRI1-associated receptor kinase 1; BIN, brassinosteroid-insensitive; BZR1, brassinazole-resistant 1; AREB/ABF, ABA-responsive element binding protein/ABRE-binding factor; PYR/RYL/RCAR, pyrabactin resistance 1/PYR-like/regulatory components of ABA receptors; PP2C2, protein phosphatase 2C; OST1, open stomata 1; PIP2, phosphatidylinositol 4,5-bisphosphate; IP3, inositol trisphosphate; CaM, calmodulin; CAMTA, calmodulin-binding transcription activator; CBL, calcineurin B-like protein; CIPK, calcineurin B-like protein; CPK, calcium-dependent protein kinases; MYB, v-myb avian myeloblastosis viral; MAPK, mitogen-activated protein kinases; SIZ1, small ubiquitin-like modifier E3 ubiquitin ligase; HOS1, high expression of osmotically responsive gene 1; BTF3, basic transcription factor 3; NO, nitric oxide; HY5, elongated hypocotyl 5; DELLA, DELLA protein; RBOH, respiratory burst oxidase homologue; ROS, reactive oxygen species; ZAT12, Zinc finger of *Arabidopsis thaliana* 12; ICE, inducer of CBF expression; CBF, C-repeat binding factor; COR, cold regulated gene.

**Figure 3 cells-14-00110-f003:**
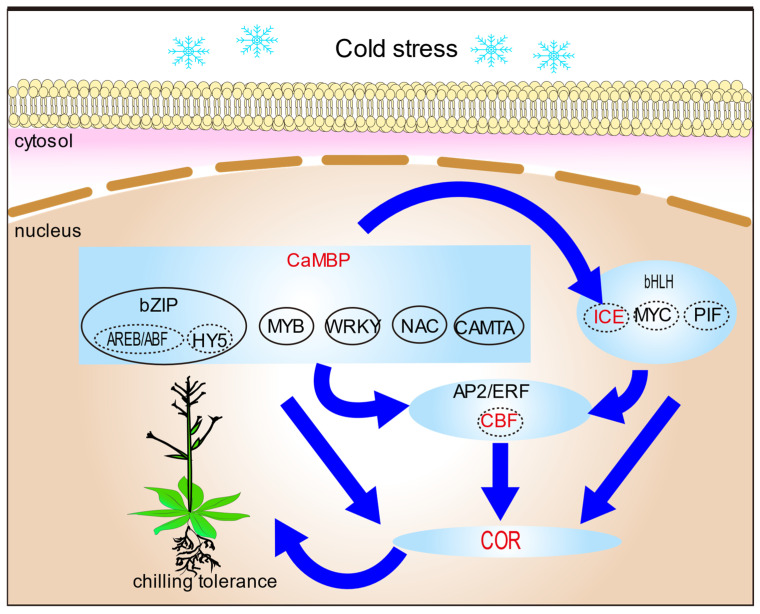
A model to explain transcription factors of cold tolerance in plants. By regulating a large number of transcription factors, the mechanisms of triggering a series of responses to plant cold tolerance can be redesigned. Abbreviations: CaMBP, CaM-binding proteins; bZIP, basic leucine zipper; AREB/ABF, ABA-responsive element binding protein/ABRE-binding factor; HY5, hypocotyl 5; MYB, v-myb avian myeloblastosis viral; WRKY, WRKY transcription factors; NAC, NAC transcription factors; CAMTA, calmodulin-binding transcription activator; AP2/ERF, aptala2/ethylene response factor; bHLH, basic helix–Loop–helix; MYC, myelocytomatosis proteins; PIF, phytochrome-interacting factor; ICE, an inducer of *CBF* expression; CBF, C-repeat binding factor; COR, cold regulated gene.

**Figure 4 cells-14-00110-f004:**
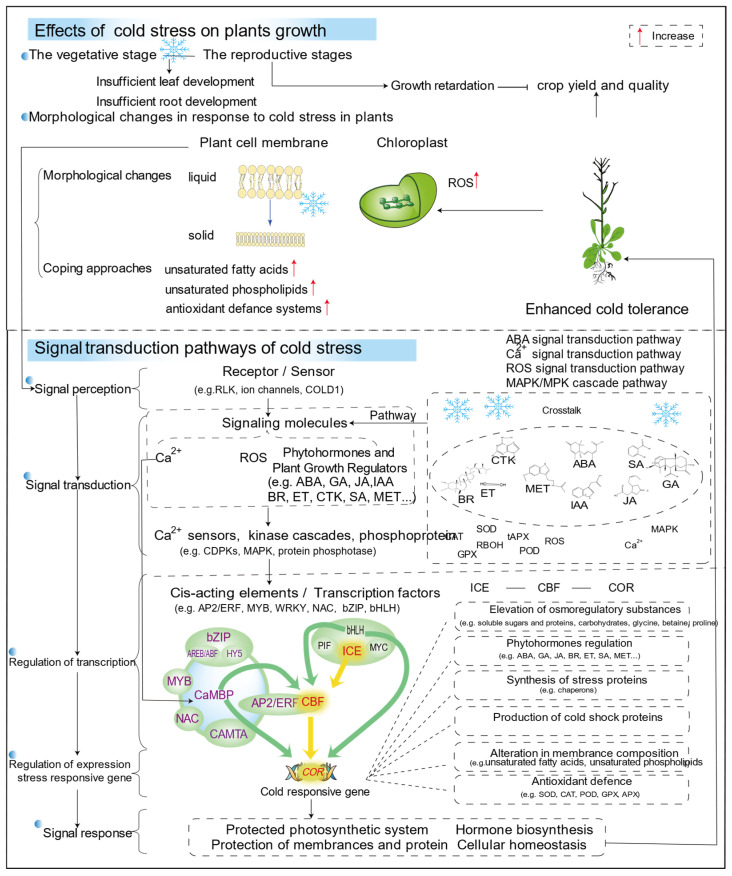
The mechanism of cold tolerance in some plants. The intercellular signal (e.g., phytohormone, osmoregulatory and inorganic ions) binds to the receptor under cold stress and is converted into an intracellular signal through the signal transduction system on the cell membrane (e.g., ion channels, RLK, and protein kinases). Cold can also be directly translated into intracellular signals. Abbreviations as above.

**Table 1 cells-14-00110-t001:** Positive and negative impacts of cold on plants.

Impacts	Class	Content	References
Positive	Cold acclimation	Improving plant cold tolerance	[18]
	Vernalization	Improvement of flowering	[19]
		Improvement of seed yield	[20]
Negative	Insufficient leaf development	Reducing leaf elongation	[21]
		Leaf chlorosis (wilting, even necrosis)	[22]
		Reducing stomatal conductance	[23]
	Insufficient root development	Swelling root tips	[24]
		Thicker root axis	[16]
		Less lateral and more seminal roots	[25]
		Reducing root length	[26]
	Growth retardation	Spikelet sterility	[27]
		Limiting seed size	[28]
		Lower survival rate	[29]

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
