# Peer review of "Plant Coping with Cold Stress: Molecular and Physiological Adaptive Mechanisms with Future Perspectives"

_cells, 2025, doi:10.3390/cells14020110_

Round 1

Reviewer 1 Report

Comments and Suggestions for Authors

Dear Authors,

Review article entitled.

Plant Coping with Cold Stress: Molecular and Physiological Adaptive Mechanisms with Future Perspectives

The present manuscript addresses an exciting and helpful topic for basic research, breeders, and breeding programs of different crops.

Cold stress is an actual topic in agriculture production in many parts of the world, and this review manuscript summarizes the current status of the knowledge very well.

The manuscript is well structured, and different sections are logically presented, which helps readers unfamiliar with the topic read and orient themselves quickly and smoothly.

The main section of this review manuscript presents the processes' complexity and the leading players, including plant growth hormones, transcription factors, and genes involved in cold response processes in plants. The readers can benefit from this work and get a very good and deep overview of this problematic. A lot of new and actual citations are also very helpful and suitable for understanding the complexity of this topic.

Abstract: well written, but some language corrections and revisions are needed.

Here is the corrected version of this section.

Abstract:

 Cold stress hinders plant growth and development, but the molecular and physiological adaptive mechanisms that help plants withstand it are poorly understood. Plants undergo several morpho-physiological changes to cope with cold stress. Plants make various changes to cope with cold stress, such as altering cellular membranes and chloroplast structure and regulating cold signals related to phytohormones (ABA, GA, JA, BR, ET, SA, and MET), reactive oxygen species  (ROS), protein kinases, and inorganic ions. This review summarizes the mechanisms of how plants respond to cold stress, covering four main signal transduction pathways, including abscisic acid  (ABA) signal transduction pathway, Ca2+ signal transduction pathway, ROS signal transduction pathway, and mitogen-activated protein kinase (MAPK/MPK) cascade pathway. Some transcription factors, such as AP2/ERF, MYB, WRKY, NAC, and bZIP, not only act as calmodulin-binding proteins during cold perception but can also play essential roles in the downstream chilling signaling pathway. This work analyzes transcription factors such as bHLH, especially bHLH-type transcription factors ICE. It discusses their functions as several phytohormones responsive elements binding proteins in the promoter region in response to cold stress.  In addition, the theory of plant response to cold stress tolerance was proposed to guide future research on the direction of agricultural production practice.

Conclusion: well written. Explaining the need for more in-depth studies of the key players and mechanisms involved in this process.

After some language corrections, I recommend this manuscript for publishing. 

23.12.2024

Author Response

Comments 1: The present manuscript addresses an exciting and helpful topic for basic research, breeders, and breeding programs of different crops.
Cold stress is an actual topic in agriculture production in many parts of the world, and this review manuscript summarizes the current status of the knowledge very well.
The manuscript is well structured, and different sections are logically presented, which helps readers unfamiliar with the topic read and orient themselves quickly and smoothly.
The main section of this review manuscript presents the processes' complexity and the leading players, including plant growth hormones, transcription factors, and genes involved in cold response processes in plants. The readers can benefit from this work and get a very good and deep overview of this problematic. A lot of new and actual citations are also very helpful and suitable for understanding the complexity of this topic.
Response 1: Thanks for your nice comments. We have improved the quality of our manuscript accordingly.

Comments 2: Abstract: well written, but some language corrections and revisions are needed.
Here is the corrected version of this section.
Abstract:
Cold stress hinders plant growth and development, but the molecular and physiological adaptive mechanisms that help plants withstand it are poorly understood. Plants undergo several morpho-physiological changes to cope with cold stress. Plants make various changes to cope with cold stress, such as altering cellular membranes and chloroplast structure and regulating cold signals related to phytohormones (ABA, GA, JA, BR, ET, SA, and MET), reactive oxygen species (ROS), protein kinases, and inorganic ions. This review summarizes the mechanisms of how plants respond to cold stress, covering four main signal transduction pathways, including abscisic acid (ABA) signal transduction pathway, Ca2+ signal transduction pathway, ROS signal transduction pathway, and mitogen-activated protein kinase (MAPK/MPK) cascade pathway. Some transcription factors, such as AP2/ERF, MYB, WRKY, NAC, and bZIP, not only act as calmodulin-binding proteins during cold perception but can also play essential roles in the downstream chilling signaling pathway. This work analyzes transcription factors such as bHLH, especially bHLH-type transcription factors ICE. It discusses their functions as several phytohormones responsive elements binding proteins in the promoter region in response to cold stress.  In addition, the theory of plant response to cold stress tolerance was proposed to guide future research on the direction of agricultural production practice.
Response 2: Thank you. We have further revised the abstract accordingly.

Abstract
Cold stress strongly hinders plant growth and development, however molecular and physiological adaptive mechanisms of cold stress tolerance in plants are not well understood. Plants adapt several morpho-physiological changes to withstand cold stress. Plants have evolved various strategies to cope with cold stress. These strategies included changes in cellular membranes and chloroplast structure, regulating cold signals related to phytohormones and plant growth regulators (ABA, JA, GA, IAA, SA, BR, ET, CTK, and MET), reactive oxygen species (ROS), protein kinases, and inorganic ions. This review summarizes the mechanisms of how plants respond to cold stress covering four main signal transduction pathways, including abscisic acid (ABA) signal transduction pathway, Ca2+ signal transduction pathway, ROS signal transduction pathway, and mitogen-activated protein kinase (MAPK/MPK) cascade pathway. Some transcription factors, such as AP2/ERF, MYB, WRKY, NAC, and bZIP, not only act as calmodulin-binding proteins during cold perception but can also play important roles in the downstream chilling signaling pathway. This review also highlights the analysis of those transcription factors such as bHLH, especially bHLH-type transcription factors ICE, and discusses their functions as several phytohormones responsive elements binding proteins in the promoter region under cold stress. In addition, a theoretical framework outlining plant responses to cold stress tolerance has been proposed. This theory aims to guide future research directions and inform agricultural production practices, ultimately enhancing crop resilience to cold stress.

Comments 3: Conclusion: well written. Explaining the need for more in-depth studies of the key players and mechanisms involved in this process.
Response 3: Thank you very much for your valuable comments. Revised portion are marked in red throughout the revised manuscript.

Comments 4: After some language corrections, I recommend this manuscript for publishing.
Response 4: We have further polished the whole manuscript. The co-author who is a native English speaker polished the manuscript. Your Suggestions have helped us to improve the deficiencies in the manuscript and increase the rigor and readability of the article. We appreciate for Editors’ warm work eamestly, and hope the correction will meet with approval. Wishing you a colorful Holiday season with your family and friends ! I wish this New Year will prove to be a happy and prosperous year for you.

Reviewer 2 Report

Comments and Suggestions for Authors

The submitted manuscript deals with the problem of low temperature effects on plants in the form of a literature review. Attention is focused on individual plant responses to this stressor. The cascades of enzymatic reactions and plant defences are described in detail. It is somewhat unfortunate that no mention is made of the formation of ice crystals within or on the surface of plant cells. It would have been useful to add, for example, information on the direction of breeding for cold/frost resistance. I therefore recommend that this be added. Literature sources are cited correctly and this is the current state of knowledge on the subject. 

Author Response

Comments: The submitted manuscript deals with the problem of low temperature effects on plants in the form of a literature review. Attention is focused on individual plant responses to this stressor. The cascades of enzymatic reactions and plant defences are described in detail. It is somewhat unfortunate that no mention is made of the formation of ice crystals within or on the surface of plant cells. It would have been useful to add, for example, information on the direction of breeding for cold/frost resistance. I therefore recommend that this be added. Literature sources are cited correctly and this is the current state of knowledge on the subject.

Response: We appreciate the valuable comments. Preventing ice crystal formation is regarded as a crucial mechanism of cold tolerance in plants. Screening and cultivation of cold tolerant germplasm resources based on strategies to prevent ice crystals is a direction of breeding for cold/frost resistance. We have added this expression according to the comment in line 115-139, 148-149 and 151-161. 

Describe as “When temperatures drop below 0 °C and fall beneath the plant's critical threshold, the formation of ice crystals within or on the surface of plant cells invariably leads to cellular damage and eventual plant death as ice crystals develop. Preventing ice crystal formation is regarded as a crucial mechanism of cold tolerance in plants. Cells may suffer mechanical damage or cellular dehydration resulting from cytoplasmic efflux due to an osmotic imbalance caused by the exclusion of solutes from recrystallized ice [28]. Downy birch (Betula pubescens Ehrh.) suffered no injury at any tissues under –70 °C cold stress [29]. Plants vary greatly in cold tolerance. It is of great significance to study the critical temperature of plants. Using USDA Plant Hardiness Zone Map, the cold tolerance of plants can be preliminarily understand [30]. The impact of cold injury on plants depends on a variety of factors, including the intensity and duration of the cold injury, the stage of plant development at which cold injury occurs, and plant characteristics such as root structure , cuticle wax composition, and cell wall thickness [31]. Trichome coverage and 3D wax projections can be recognized as antifreezing strategies of plants, which increasing the thickness of the cuticle layer, stomatal density, and cuticular permeability [31]. Plants respond to cold stress through a series of physiological and morphological changes, such as ice-binding proteins and antifreezing proteins [28]. The ice-binding microalgal protein CmIBP1 effectively inhibits ice recrystallization both in vitro and in transgenic plants, as the intensity of IRI activity can be compared based on the size of the ice particles at a given temperature and duration. [32]. CmIBP1 is secreted into apoplasts and improve freezing tolerance by inhibiting the growth of ice crystals in transgenic plants [32]. The reason for ice nucleation and freezing in plants are unknown. Studying cold perception in plants can provide the direction of ice diffusion from nucleation site. Screening and cultivation of cold tolerant germplasm resources based on strategies to prevent ice crystals, balancing the effects of growth and defense.”in line 115-139.

“Ice crystals’ formation can increase electrolyte leakage and causes lipid peroxidation [35].” in line 115-139. 

“Studies have shown that a specific haplotype of the OsSRO1c gene (OsSRO1cHap1) can significantly improve rice (Oryza sativa L.) a cold tolerance at both seedling and heading stages. Further biochemical analysis shown that the OsSRO1c protein exhibits liquid-liquid separation characteristics properties in the nucleus and form a condensate with OsDREB2B. These condensates can directly respond to cold stress and form small droplet structures through protein phase transformation [38]. The strong haplotype1 encoding the OsSRO1c protein d showed better mobility in condensed droplets formed by OsDREB2B and increased the transcriptional activity of OsDREB2B, thereby activating the first plant low temperature sensors, including COLD1 [38] (Figure 1). This mechanism enables O. sativa with enhanced cold tolerance.”in line 151-161.

Reviewer 3 Report

Comments and Suggestions for Authors

The paper is written in a very general way. The best part is the part devoted to transcription factors. The text is based almost entirely on studies on Arabidopsis, there are very few references to other plants, including cultivated ones. I suggest that this fact should be included in the title and abstract of the paper. The paper needs to be unified in terms of editing, because individual fragments were probably written by different authors and this is visible. 

Detailed comments are indicated in the manuscript.

Author Response

Comments 1: The paper is written in a very general way. The best part is the part devoted to transcription factors. The text is based almost entirely on studies on Arabidopsis, there are very few references to other plants, including cultivated ones. I suggest that this fact should be included in the title and abstract of the paper. The paper needs to be unified in terms of editing, because individual fragments were probably written by different authors and this is visible.
Response 1: Thank you for pointing out these problems. We have revised the whole paper according to your suggestions. Cold tolerance of plant is a complex trait controlled by multiple genes, involving many gene regulatory pathways, and cross-interacting with other environmental stress. Phytohormones play a crucial role in regulating plant cold stress tolerance. Phytohormone regulatory mechanism of plants under cold stress is poorly known. As a molecular switch, transcription factors can bind to cis-acting elements in gene promoter region to directly regulate the expression of downstream functional genes, and can also affect the expression of a series of downstream functional genes by regulating the expression of other transcription factors. So using biotechnology to overexpress or silence a specific transcription factor in the plant, they could improve the plant's cold stress resistance. In-depth exploration of phytohormone synthesis, decomposition, and gene expression changes related to signal transduction under cold stress will greatly enrich the analysis of plant cold resistance. We added some reference and Latin name. 59 of 208 articles studies on Arabidopsis. Among many phytohormones, ABA can act as a signal molecule together with many other phytohormones when plants respond to cold stress. So, in the first step, we introduced ABA. For the editing of the figure 2, the order of phytohormones in the text is inconsistent with that in the figure2. Thanks for your careful checks. We are sorry for our carelessness. Based on your comments, we have made the corrections to add the IAA and CTK sections. 

Comments 2: “various stages of plant growth” in line 61, what do you mean?
Response 2: Thanks for your comments. It was typographical error. We have corrected it in the revised version. “cold stress at all stages of plant growth and development should be paid attention to.”.

Comments 3: “Current research on cold tolerance has focused on the germination and emergence stages, with relatively little research on the reproductive stage.” in line 65-66, I do not agree.
Response 3: Failure to express clearly is our mistake in writing. We would like to express that it is meaningful to carry out research on cold stress in plants at different periods. The same plant tissue (e.g. leaf and root) can be affected by different degrees of cold stress at different growth stages. We have modified this expression according to the comment in line 66-68. Describe as “All stages of plant development, from seed germination, flowering, and fruiting to dormancy, are impacted by cold stress. Various levels of cold stress can influence the same plant tissue at distinct growth stages.”.

Comments 4: “15 hybrids (45%) had” in line 69, means nothing
Response 4: We thank the reviewer for the reminder. We would like to express that root can be an indicator for identifying and evaluating the cold tolerance of plant. We have added this expression according to the comment in line 70-72. Describe as “Through principal component analysis, root characteristics may aid in identifying corn (Zea mays L.) hybrids that exhibit cold tolerance [16].”.

Comments 5: “cell yield” in line 71, what do you mean?
Response 5: Thank you for pointing out this problem. We have added this expression according to the comment in line 73-79. Describe as “Kinematic analysis of diurnal growth rates in control and cold-treated corn leaves from germination until the completion of leaf 4 expansion showed that cold nights had an im-pact on both cell cycle time (+65%) and cell yield (-22%), meanwhile the size of mature epidermal cells was unaffected. This finding contrasts with the common belief that the reduction in growth caused by abiotic factors is typically attributed to a combination of decreased cell production and reduced mature cell size [17].”.

Comments 6: “It reduces seedling growth and biomass production by increasing” in line 72, it is simplification.
Response 6: We are grateful for the reviewer's suggestion. We have incorporated additional details to show that the cold tolerance of plant is a complex trait. We have added the suggested content to the manuscript in line 80-85. Describe as “Plants undergo a series of physiological and morphological changes to cope with cold stress including increasing malondialdehyde (MDA) content, membrane permeability, proline accumulation, and altering antioxidant enzymes including superoxide dismutase (SOD), ascorbate peroxidase (APX), catalase (CAT), peroxidase (POD) and glutathione peroxidase (GPX) activities, such effects are generally achieved through significant transcriptional regulation”.

Comments 7: “Prolonged cold stress through early planting has been found to induce cellular membrane damage and growth retardation in rice crops via exposing rice to cold through early planting ” in line 76-78, repetition early planting.
Response 7: Thank you for pointing this out. We have modified this expression according to the comment in line 87-88. Describe as “The influence of cold stress during early growth stages can significantly affect later growth phases.”.

Comments 8: “via the vernalization” in line 79, vernalization does not refer to seed germination, maybe you were thinking about cold stratification.
Response 8: Thanks. Plants can enhance seed germination through cold stratification. We have re-written this part according to the reviewer's suggestion in line 92-95. Describe as “Plants can enhance seed germination through cold stratification [23]. Experiences during one developmental stage can leave lasting impacts on later stages. Notably in in arabidopsis (A. thaliana), rosette vernalization has been shown to boost seed germination across various ecotypes [24].”.

Comments 9: “early developmental phase of wheat causes” in line 82, specify
Response 9: We have added the suggested content to the manuscript in line 97-101. Describe as “with each extra day of freezing at the critical temperature, plant mortality rates increased by 8.6%, 22.3%, 11.1%, and 9.4% for the wheat (Triticum aestivum L.) cultivars Jing411, Nongda211, Zhengmai366, and Yanzhan4110, respectively. Among the same cultivar, tillers demonstrated lower sensitivity to freezing duration (15–32% per day) when compared to younger leaves (25–35% per day) and older leaves (20–55% per day) [27]”.

Comments 10: “In summary, cold stress disrupts plant cell membranes, suppressing mineral nutrient uptake, biosynthesis, and light energy uptake, blocking physiological functions. This can lead to permanent tissue damage and plant death in severe cases” in line 106-109, this part can be omitted. It is the repetition
Response 10: It is simplification. We have deleted it.

Comments 11: “Figure 1” in line 111, The figure is not informative.
Response 11: We think this is an excellent suggestion. We have corrected the title of Figure 1 “The perception of cold stress on plant” into ”Cold stress impact on plants.” in line 163. We would like to express that cold damage occurs when the cell membrane transforms from a fluid state to a solid state. Plant temperature sensors, including COLD1, and Chloroplast are affected at low-temperature.

Comments 12: “Vernalization” in line118, Vernalization does not refer to seed germination - please modify.
Response 12: We sincerely thank the reviewer for careful reading. We have corrected the title of Table 1 “Benefits and problems of cold on plants.” into ” Positive and negative impacts of cold on plants.”, added the part of “Cold stratification” in Table 1.

Comments 13: “maintain PSII performance under cold stress, plants modify chloroplast structure” in line 131-132, so wats going on, explain in details?
Response 13: We sincerely appreciate the valuable comments. We have checked the literature carefully and explained in details in line 186-192. Describe as “he seasonal movement of chloroplasts is mainly affected by low temperature stress. Under greenhouse conditions close to natural light and photoperiod, chloroplasts maintains their activity in the upper plate [49]. The timing of cytoplasmic cluster formation in the two Picea species studied coincides with the minimum seasonal level of chlorophyll fluorescence parameter characterizing the efficiency of the photosynthetic apparatus [49]. This period also corresponds to a decrease in grana development within the chloroplasts [49].”.

Comments 14: “Plant cold response depends on light quality, as cold exposure may induce PSII photoinhibition and oxidative damage” in line 133-135, explain why, it is simplification, and gives no information.
Response 14: We think this is an excellent suggestion. We have explained in details in line 195-205. Describe as “Plants produce excess energy beyond their utilization capacity under stress [18]. This surplus energy leads to a decrease in the photosynthetic rate and electron transport capacity, resulting in photoinhibition. Although CA treatment can significantly mitigate the degree of PSII photoinhibition and oxidative damage in tobacco (Nicotiana tabacum L.) leaves under cold stress [18]. Seedlings can adapt to CA by regulating energy dissipation, thereby preventing excessive reduction in plastoquinone pools and subsequent PSII photoinhibition [18]. However, severe stress may progressively exacerbate PSII photoinhibition, if the excess excitation energy is not adequately dissipated through non-photochemical quenching pathways and other electron sinks in a timely manner. This further leading to increased production of ROS generation, thus damaging the photosynthetic mechanism [18].”。

Comments 15: “through exogenous substances” in line147, do you mean by treatent with such substances? Phytohormones, ROS etc are endogenous chemicals...
Response 15: We were really sorry for our careless mistakes. Thank you for your reminder. We have modified this expression according to the comment in line 218-219. Describe as “through phytohormones, plant growth regulators, ROS, protein kinases, and Ca2+.”。

Comments 16: “Phytohormones” in line 150, In the whole chapter I see the lack of examples,. Authors present only general information abauts signal stransduction pathways.
Response 16: We sincerely appreciate the valuable comments. We have checked the literature carefully and added more references on and into this part in there vised manuscript. As suggested by the reviewer, we have corrected the “Phytohormones" into “Phytohormones and Plant Growth Regulators", and added IAA and CTK parts.

Comments 17: “ICE-CBF-COR” in line 166, explaine all abbreviations.
Response 17: Thank you for your reminder. We have modified in line 236-237. Describe as “the inducer of CBF expression (ICE) - C-repeat binding factor (CBF) - cold regulated gene(COR)”.

Comments 18: “See the text for additional details.” in line 175, In the text there are no details to all pathways.
Response 18: We have carefully checked the manuscript and corrected the errors accordingly. As suggested by the reviewer, we have added IAA and CTK parts.

Comments 19: “a hub between ABA, SA, GAs, and IAA signaling” in line 231, Are there any proofs? Include citations
Response 19: We have added the reference to support this idea in line 297-301. Describe as “a hub between ABA, SA, GAs, and IAA signaling [62]. MYC2 positively and negation regulates multiple functions in the JA signaling pathway [62]. Physical interactions with other key regulatory proteins, formation of heterodimers with other transcription factors, and the ability to activate or repress gene expression in response to multiple signals can contribute to the diverse regulatory roles of MYC2 [62].”.

Comments 20: “CBF1 and CBF2” in line 240, should be in italics
Response 20:. Thanks for your careful checks. We are sorry for our carelessness.As suggested by the reviewer, we have corrected the “CBF1 and CBF2 " into “CBF1 and CBF2” in line 356.

Comments 21: “Melation (MET)” in line 254, Melatonin is not included into a group of phytohormones, it is a plant growth regulator.
Response 21:. We sincerely appreciate the valuable comments. Melatonin is not included into a group of phytohormones, it is a plant growth regulator. As suggested by the reviewer, we have corrected the “Phytohormones " into “Phytohormones and Plant Growth Regulators".

Comments 22: “MT” in line 260, Do you mean melatonin? Use the proper abbreviation MET.
Response 22: Thanks for your careful checks. We are sorry for our carelessness. As suggested by the reviewer, we have corrected the “MT " into“MET" in line 418. 

Comments 23: “Non-Phytohormones” in line 263, Signalling compounds
Response 23:. We think this is an excellent suggestion. As suggested by the reviewer, we have corrected the “Non-Phytohormones" into“Signalling compounds" in line 421.

Comments 24: “sAPX and tAPX trigger expression of COR15A, PAL1, and CHS” in line 268-269. unclear.
Response 24: We sincerely appreciate the valuable comments. We have checked the literature carefully and explained in details in line 428-435. Describe as “Long-term exposure to cold stress showed that sAPX was not relevant and showed a strong dependence on tAPX [98]. Thylakoid protection mediated by tAPX acts as an initia-tion center that stores initiation information over time [99]. tAPX-mediated thylakoid pro-tection serves as a priming hub, which stores information on priming over time [99]. Compared with plants induced by short-term cold stress, long-term cold stress induced stronger induction of non-chloroplast-specific ROS-regulated genes such as CHS and PAL1 (and COR15A), which support salicylic acid, lignin, flavonoids, and Floral biosyn-thesis of various secondary stress protective mediators such as anthocyaninsg [99].”.

Comments 25: “AtXTH21 [67]. Overexpression of the ROS signal response gene AtZAT12 downregulates AtCBF1/2/3 genes” in line 271. Such information are completly incoprehensible.
Response 25: We have checked the literature carefully and explained in details in line 437-442, and 444-448. Describe as “Genetic evidence indicated that AtHAP5A acts upstream of AtXTH21 in freezing stress response in A. thaliana [100]. These results revealed that AtHAP5A modulates freezing stress resistance through interaction with the CCAAT motif of AtXTH21 in A. thaliana [100]. Overexpressing the AtHAP5A and AtXTH21 could alleviate 4°C stress-induced ROS accumulation and related oxidative damage in A. thaliana, while AtHAP5A and AtXTH21 mutants had the opposite effect [100].”. “ZAT12 downregulated the expression of the CBF genes indicating a role for ZAT12 in a negative regulatory circuit that dampens expression of the CBF cold response pathway [101]. The role of the ZAT12 regulation may help plants cope with oxidative stress. In this regard, it is of great interest that ZAT12 expression resulted in downregulation of transcripts encoding a putative l-ascorbate oxidase [101].”.

Comments 26: “Cold-resistant plants respond quickly to cold stress, with positive gene responses to reduce ROS and cope with damage” in line 274-275.
Response 26: We have deleted it.

Comments 27: “plant cold induction” in line 335. what do you mean?
Response 27: in line 511-513. Describe as “Hypothermia receptors mostly trigger calcium ions. It is important to create a suitable expression pattern for positive regulators of cold resistance without negatively affecting favorable agronomic traits.”

Comments 28: “Aptala2/ethylene response factor (AP2/ERF) is the most studied transcription factor involved in the mechanism of cold response” in line 347-348. It gives no interesting information.
Response 28: It is a simple introduction for better reading. If you think it is unnecessary, we can delete it.

Comments 29: “C-repeat binding factor/ Dehydration response element binding factor 1 (CBF/DREB1) belongs to its subfamily, and there are many studies on its regulation under cold stress” in line 358-359. What is the importance of such information?
Response 29: CBFs belong to plant transcription factor AP2/ERF protein family. ICE-CBF-COR is the main cold stress research pathway.

Comments 30: “ICE-CBF-COR is the main cold stress research pathway.” in line 358-359. This is repeated several times in the text
Response 30: We have deleted it.

Comments 31: “CBF transcription factors are also regulated by hormones. Future research should focus on exploring more cold-inducing genes and the link between hormone response and downstream signaling factors.” in line 361-363. This is inconsistent with the data in the figure 2.
Response 31: We have deleted it to avoid misunderstanding. ICE-CBF-COR is the main cold stress research pathway. At present, these studies focus on the function of a single gene, most of which are CBF-dependent pathways. However, the cold response transcriptional group regulated by CBFs accounted for only about 12%. Exploring more cold-inducing genes and the link between hormone response and downstream signaling factors is also important.

Comments 32: “pepper” in line 370. Include Latin name
Response 32: Thanks for your careful checks. We have modified this expression according to the comment in line 539. Describe as “pepper (Capsicum annuum L.)”.

Comments 33: “BMY” in line 374. use italics
Response 33: Your suggestion really means a lot to us. We did mistaked BYM to α-amylase 1a (OsAMY1a). We have modified this expression according to the comment in line 553. Describe as “OsAMY1a

Comments 34: “maltose content” in line 375. What is the change in maltose content?
Response 34: Your suggestion really means a lot to us. We mistaked maltose content to trehalose. We have modified this expression according to the comment in line 554-559. Describe as “OsMYB30-OsTPP1 is a sugar signaling pathway that regulates seed germination in response to low temperature. Expression of OsMYB30 and OsTPP1 was induced by low temperature during seed germination [146]. OsMYB30 binds to the promoter region of OsTPP1 to activate its expression [146]. Overaccumulation of trehalose was found in both OsMYB30- and OsTPP1-overexpressing lines, resulting in inhibition of OsAMY1a during seed germination [146]”

Comments 35: “affects chlorophyll and anthocyanin” in line 376-377. specify what does it mean affect?
Response 35: We have checked the literature carefully and explained in details in line 559-561. Describe as “The CaMYB306 transcription factor inhibits the positive cold resistance regulator CaCIPK13 in C. annuum. It also suppresses chlorophyll and anthocyanin contents and regulates ROS signaling”.

Comments 36: “MYB transcription factors have potential in plant breeding and improvement, but their underlying mechanisms remain unclear. Some genes may have adverse effects on growth and development, limiting their direct use for enhancing resistance. Exploring MYB genes and using new methods like tissue-specific expression are crucial for improving crop resistance and achieving genetic progress.” in line 377-382. Only general information, it does not carry any knowledge
Response 36: We have deleted it.

Comments 37: “Ipomoea batatas” in line 385. italics, include also common name of the plant
Response 37: Thanks for your careful checks. We have modified this expression according to the comment in line 564. Describe as ”sweat potato (Ipomoea batatas (L.) Poir.)”.

Comments 38: “tobacco” in line 405. Include Latin name of the plant.
Response 38: Thanks for your careful checks. We have modified this expression according to the comment in line 584-586. Describe as “Chorispora bungeana (Chorispora bungeana Fisch. et Mey.) CbABF1 confers stress tolerance in N. tabacum by reducing ROS and enhancing antioxidant enzymes”.

Comments 39: “rice” in line 410. If appeared the first time in the text include also Latin name of the plant.
Response 39: We have appeared the first time in the text include also Latin name of the plant. Describe as “rice (Oryza sativa L.)” in line 90, and “O. sativa” in line 591.

Comments 40: “reactive oxygen species” in line 414. ROS.
Response 40: Thanks for your careful checks. We have modified this expression according to the comment in line 595. Describe as “ROS”.

Comments 41: “Arabidopsis thaliana” in line 419. Uniform all over the text
Response 41: We have appeared the first time in the text include also Latin name of the plant. Describe as “arabidopsis (A. thaliana (L.) Heynh.)” in line 153, and “A. thaliana” in line 599-600.

Comments 42: “Mutants” in line 425. small letter
Response 42: Thanks for your careful checks. We have modified this expression according to the comment in line 606. Describe as “mutant”.

Comments 43: “Poncirus trifoliata” in line 459. Include common name of the plant, the proper name of the plant is P. trifoliata (L.) Raf. Use the proper names of the plant all over the text..
Response 43: We have appeared the first time in the text include also Latin name of the plant. Describe as “trifoliate orange (Poncirus trifoliata (L.) Raf.)” in line 643-644.

Comments 44: “plants” in line 463. Use a proper Latin names of the plants
Response 44: Thanks for your careful checks. We have modified this expression according to the comment in Table 2.

Comments 45: “Glutathione(GSH)” in line 487. Don't use the full name and abbreviation instead. If you enter an abbreviation, use it all over the text.
Response 45: Thanks for your careful checks. We have modified this expression according to the comment in line 672. Describe as “Glutathione”.

Comments 46: “tomato” in line 496. Include Latin name.
Response 46: We have modified this expression according to the comment in line 678. Describe as “S. lycopersicum SlWRKY”.

Comments 47: “Proteomic data showed” in line 496. include information on the plant the experiement was performed
Response 47: We have modified this expression according to the comment in line 681-683. Describe as “1-month-old A. thaliana exposed for 1 week to 4°C at short-day conditions under white (100 and 20 μmol m−2s−1), blue, or red (20 μmol m−2s−1) light conditions [201].”.

Comments 48: “sugarbeet” in line 499. Include Latin name of the plant.
Response 48: Thanks. We have done as as “sugarbeet (Beta vulgaris L.)” in line 686.

Comments 49: “improved glycans and amino acid metabolism” in line 502. I do not understand ... the plant.
Response 49: We are very sorry for the deviation in writing and caused the reviewer’s misunderstand. We have modified this expression according to the comment in line 688-691. Describe as “Advances in plant multi-omics, such as genomics, transcriptomics, prote-omics, microbial functional genomics, epigenetics, and metabolomics, have accelerated the discovery of the molecular mechanisms behind plant cold tolerance”.

Comments 50: “discussed the research progress of plant performance” in line 505. Authors presented only few data reffering to other plants than Arabidopsis, so the whole text is based on this model plant - I do not agree such point of view. If so, Arabidopsis should be included in the title of the manuscript.
Response 50: Thank you for pointing out these problems. We have modified this expression according to the comment in line 693-694. Describe as “In conclusion, this review discusses research progress of some plants coping with cold stress, and reg-ulatory mechanism of cold tolerance in plants.”We have redrawn the Figure 4. The cold tolerance of plant is a complex trait controlled by multiple genes, involving many gene regulatory pathways, and cross-interacting with other environmental stress. Phytohormones play a crucial role in regulating plant cold tolerance. Phytohormone regulatory mechanism of plants under cold stress is still largely a mystery. As a molecular switch, transcription factors can bind to cis-acting elements in gene promoter region to directly regulate the expression of downstream functional genes, and can also affect the expression of a series of downstream functional genes by regulating the expression of other transcription factors. So using biotechnology to overexpress or silence a specific transcription factor in the plant, they could improve the plant's cold stress resistance. In-depth exploration of phytohormone synthesis, decomposition, and gene expression changes related to signal transduction under cold stress will greatly enrich the analysis of plant cold resistance. We added some reference and Latin name. 59 of 208 articles studies on Arabidopsis. Please let me know if there is anything need to be revised. If there are any other modifications we could make, we would like very much to modify them and we really appreciate your help.

Comments 51: “Figure 4” in line 518. there is a mistake in the words: phytohormone, osmoregulatory, do not include MET into a group of hormones. What do you mean by the phrase "first messenger" is it a primary factor, messenger?
Response 51: Thank you for pointing this out. We were really sorry for our careless mistakes. The reviewer is correct. we have corrected the “phytohormone, osmotic regulatotion" into “phytohormone, osmoregulatory”, and MET into hormones growth regulator. It is a primary factor, messenger.

Comments 52: “Malus baccata” in line 534. include common name of the plant, the proper Latin name is Malus baccata Borkh. - modify
Response 52: We have appeared the first time in the text include also Latin name of the plant. Describe as “malus baccata (Malus baccata Borkh)” in line 614, and ““M. baccata  Borkh.”.in line 723. 

Comments 53: “cotton” in line 534. include Latin name
Response 53: We have appeared the first time in the text include also Latin name of the plant. Describe as “cotton (Gossypium hirsutum L.)” in line 725-726. 

Thank you very much for your valuable comments. Your Suggestions have helped us to improve the deficiencies in the manuscript and increase the rigor and readability of the article. We appreciate for reviewer’s warm work eamestly, and hope the correction will meet with approval. 
Wishing you a colorful Holiday season with your family and friends ! I wish this New Year will prove to be a happy and prosperous year for you.

Round 2

Reviewer 3 Report

Comments and Suggestions for Authors

The manuscript was improved according to all my suggestions